# Conformalized Time Series with Semantic Features

**Baiting Chen**
Department of Statistics and Data Science
UCLA
brantchen@g.ucla.edu

**Zhimei Ren**
Department of Statistics and Data Science
University of Pennsylvania
zren@wharton.upenn.edu

**Lu Cheng**
Department of Computer Science
University of Illinois Chicago
lucheng@uic.edu

## Abstract

Conformal prediction is a powerful tool for uncertainty quantification, but its application to time-series data is constrained by the violation of the exchangeability assumption. Current solutions for time-series prediction typically operate in the output space and rely on manually selected weights to address distribution drift, leading to overly conservative predictions. To enable dynamic weight learning in the semantically rich latent space, we introduce a novel approach called Conformalized Time Series with Semantic Features (CT-SSF). CT-SSF utilizes the inductive bias in deep representation learning to dynamically adjust weights, prioritizing semantic features relevant to the current prediction. Theoretically, we show that CT-SSF surpasses previous methods defined in the output space. Experiments on synthetic and benchmark datasets demonstrate that CT-SSF significantly outperforms existing state-of-the-art (SOTA) conformal prediction techniques in terms of prediction efficiency while maintaining a valid coverage guarantee.

## 1 Introduction

Uncertainty quantification is essential for reliable predictions in time series data [34, 45]. The emergence of 'black-box' models has intensified interest in conformal prediction (CP), a technique valued for its model-agnostic and distribution-free properties [44, 57]. Under the assumption of data exchangeability, the confidence bands provided by CP are theoretically guaranteed [43]. This reliability has led to CP's promising performance across various domains, e.g., image classification [1], graph neural networks [56], anomaly detection [22], and natural language processing [37, 16].

The fundamental assumption of exchangeability in CP is often compromised in time series data due to inherent temporal dependencies, as highlighted in studies such as [53] and [48]. This violation poses significant challenges in directly applying CP to time series forecasting. Existing adaptations of CP tailored for time series (e.g., [7]) tend to produce overly conservative prediction sets or intervals too wide to provide practical insight, thus limiting their practical utility. These adaptations often necessitate manually selected weights, which curtail their generalizability across various datasets. Moreover, the practice of calculating non-conformity scores based on output space does not take full advantage of the rich representational data available in the latent space. This oversight is particularly critical given the prevalent use of neural networks (NN) in contemporary time series forecasting, which are capable of capturing deep, complex patterns in data as discussed in [55] and [39].

38th Conference on Neural Information Processing Systems (NeurIPS 2024).

The latent space in NN offers enhanced model interpretability and the ability to capture complex, nonlinear relationships in the data that are often invisible in the original input space ([40]). By mapping input data into a latent feature space through the feature function, the resulting representation not only supports the model in learning essential temporal patterns but also in generalizing better to unseen data by focusing on the most informative features. To overcome the inherent challenges in applying CP to time series data, our work presents a novel approach—*Conformalized Time Series with Semantic Features (CT-SSF)*—which leverages latent semantic features of NN for time series. Central to our methodology is the use of weighted errors for constructing non-conformity scores in the latent space, drawing upon rich semantic features. The weights are dynamically adjusted to prioritize semantic features more pertinent to the current prediction task. This approach ensures that the weights adapt to the changing importance of different data aspects over time. By shifting CP into the latent space of NN, our non-conformity scores achieve a deeper insight into the temporal dynamics. This not only maintains the requisite coverage of predictive intervals but also significantly improves their prediction efficiency, offering a refined tool for time series forecasting that aligns with the complexities of modern data structures.

Our contributions are summarized as follows:

- We propose CP in the latent space of the underlying time-series prediction model, enabling the construction of non-conformity scores that encapsulate a more comprehensive understanding of the data and deliver tight prediction sets.

- We propose an adaptive weight adjustment scheme to refine the weight learning process in the latent space, accounting for distribution shifts in time-series data by emphasizing data points more relevant to the current time step.

- We conduct extensive experiments under both synthetic and realistic settings to corroborate the effectiveness of the proposed algorithm, showing a 10%-20% improvement over the SOTA approach for real-world data.

## 2 Related Work

**CP with exchangeable data** CP, pioneered by [43], has become a cornerstone in uncertainty quantification due to its model-free and distribution-free property [35]. The work in this domain can be summarized of CP into two branches: improve the efficiency of CP ([33, 13, 51, 36, 6, 27] ) and generalize CP to different settings, such as quantile regression [32], decision tree [20], random forest [21]), survival analysis [11], k-nearest neighbor [30], online learning [8] and auction design [17]. Comprehensive insights into CP and its theoretical underpinnings are provided by [4].

**Beyond Exchangeability** The foundational works of [44] and the introductory reviews by [4] describe the evolution of CP to accommodate contexts "beyond exchangeability". Notable extensions include [41] application to covariate shift, and innovative adaptations for dealing with label shift by [31]. The work of [41] and [7] address calibration and test set shifts by reweighting data points. [25] show how reweighting can be extended to causal inference setups for predictive inference on individual treatment effects, and [11] show how to apply these ideas in the context of censored outcomes in survival analysis.

**CP for time series data** CP for time series can be divided into two primary trends that deviate from the traditional assumption of exchangeability. The first trend focuses on adaptively adjusting the error rate $\alpha$ during the testing phase to enhance coverage accuracy. This approach is demonstrated in works such as ([2, 3, 8, 14, 53, 26, 46, 52, 15, 38, 58]). The second trend (e.g., [41, 48, 49, 7, 23, 24, 47]) emphasizes manually assigning weights to historical non-conformity scores, giving greater importance to those that reflect the current scenario more accurately. To address the limitation of manual selection, the advanced HopCPT model, developed by [5], leverages the Modern Hopfield Network to capture temporal structure and learn the weights. The work closest to ours is the HopCPT ([5]). We advance HopCPT in developing the non-conformity score in the latent space with rich semantic features, enabling a deeper analysis and utilization of the inherent temporal patterns in the data. Alternative probabilistic frameworks like Mixture Density Networks [10] and Gaussian Processes [9] offer robust methodologies for time series forecasting beyond CP. However, these methods frequently grapple with computational limitations and lack robust theoretical backing.

## 3 Background

**Split Conformal Prediction** One of the most common approaches in CP is the *split conformal prediction* method [29], which starts with a trained model and assesses its efficacy on a calibration set of paired examples $\{(X_i, Y_i)\}_{i=1}^n$. Central to split CP is the foundational assumption that the data are exchangeable, ensuring that the errors observed in the test set will be consistent with those from the calibration set. This allows for reliable empirical estimation of quantiles across both datasets. Define a non-conformity score function $s(x, y)$ that quantifies the disagreement between predicted and ground-truth values. The coverage guarantee of CP is defined as follows:

$$P(Y_{n+1} \in C(X_{n+1})) \geq 1 - \alpha, \quad C(X_{n+1}) = \{y : s(X_{n+1}, y) \leq \hat{q}\}. \tag{1}$$

where $(X_{n+1}, Y_{n+1})$ is a new data point, $\hat{q}$ is the $\lceil(1 - \alpha)(n + 1)\rceil$-th smallest elements among $\{s(X_i, Y_i)\}_{i=1}^n$ and $\alpha$ signifies a predefined miscoverage rate.

**Non-exchangeable Conformal Prediction** Real-world settings encounter challenges such as data drift and inter-dependencies, necessitating further adaptations of CP to these non-exchangeable settings. One common solution is to use reweighting [7, 5]. For example, given a set of pre-specified weights $\{w_i\}_{i=1}^n$, $w_i \in [0, 1]$, the coverage guarantee for seminal work NexCP [7] is articulated as follows:

$$P(Y_{n+1} \in C(X_{n+1})) \geq 1 - \alpha - \sum_{i=1}^n \tilde{w}_i d_{\text{TV}}(Z, Z^i), \quad \tilde{w}_i = \frac{w_i}{1 + \sum_{i=1}^N w_i}, \tag{2}$$

where $Z = (X_1, Y_1), \ldots, (X_n, Y_n), (X_{n+1}, Y_{n+1})$ represents a sequence of $n$ calibration examples along with a subsequent test example and $\tilde{w}_i$ is the normalized weight. The term $Z^i$ denotes the sequence $Z$ after the $i$-th pair $(X_i, Y_i)$ is swapped with the test example $(X_{n+1}, Y_{n+1})$, and $d_{\text{TV}}(Z, Z^i)$ quantifies the dissimilarity introduced by this swap.

To construct prediction sets, NexCP determines the quantile threshold $\hat{q}$ by the following equation:

$$\hat{q} = \inf \left\{ q : \sum_{i=1}^n \tilde{w}_i 1\{s_i \leq q\} \geq 1 - \alpha \right\}. \tag{3}$$

Eq. 3 is consistent with the standard CP when all weights $\{\tilde{w}_i\}_{i=1}^n = 1$. Intuitively, closer alignment to exchangeable data minimizes the calibration terms $d_{\text{TV}}(Z, Z^i)$, allowing for more precise calibration. Strategic weight allocation—such as assigning larger weights to calibration points $(x_i, y_i)$ that have similar distributions in $Z$ and $Z^i$ and smaller weights where distributions diverge—can yield tighter bounds on the prediction sets. For time series data, this suggests that greater weights should be assigned to more recent observations to reflect their increased relevance.

## 4 Methodology

In this section, we first formally define CP for time-series prediction and then propose our framework that leverages rich semantic features in latent space modeling and employs a dynamic weighting mechanism that adjusts to temporal dependencies based on these semantic features, simultaneously achieving valid coverage and high efficiency.

### 4.1 Problem Definition

Given a multivariate time series $\mathbf{Z}_t = (\mathbf{x}_t, y_t)$ for $t = 1, \ldots, T$, where each $\mathbf{x}_t \in \mathbb{R}^m$ is a feature vector with $m$ dimensions and $y_t$ is the corresponding real-valued target. The prediction model $\mu$ utilizes these feature vectors to produce point predictions $\hat{y}_t = \mu(\mathbf{x}_t)$. The goal of CP is to construct a prediction interval $C_\alpha^t(\mathbf{x}_{t+1})$ that contain the true value $y_{t+1}$ with a confidence level of $1 - \alpha$:

$$P(Y_{t+1} \in C_\alpha^t(\mathbf{x}_{t+1})) \geq 1 - \alpha, \tag{4}$$

where the probability is over the randomness of $\{\mathbf{Z}_s\}_{s \leq t}$ and $\mathbf{x}_{t+1}$. This interval aims to balance reliability with informativeness—beyond ensuring the coverage, it is ideal that the prediction interval is of short length.

A common choice of non-conformity score is the absolute errors $|y_t - \mu(\mathbf{x}_t)|$ between observed values and the model's predictions [43, 32]. The prediction interval is then calculated based on the empirical $1 - \alpha$ quantile $Q_{1-\alpha}$ of the non-conformity scores:

$$C_\alpha^t(\mathbf{x}_{t+1}) = \mu(\mathbf{x}_{t+1}) \pm Q_{1-\alpha}\left(\{|y_i - \mu(\mathbf{x}_i)|\}_{i=1}^t \cup \{\infty\}\right). \tag{5}$$

## 4.2 Non-conformity Score in the Latent Space

Given the rich semantic information in the latent space, we investigate the effectiveness of using semantic features for CP with time series data. Our base model architecture $\mu$ is a deep neural network (e.g., RNN) which is structured around two key sub-neural networks comprised of linear layers $\mu = g \cdot f$: The feature function $f$ maps input data into a latent feature space, while the prediction head $g$ transforms these features into the forecasted outputs.

In typical supervised learning for time series, ground truth labels are available only in the output space. The latent space, by contrast, is designed to capture abstract, non-obvious patterns in the data. These features help the model's learning but do not correspond directly to observable labels ([19]). To address the lack of ground-truth labels in the latent space, we apply the surrogate feature [40] to replace the ground-truth term when constructing the non-conformity score. Concretely, we define the nonconformity score to be

$$s(X, Y, \hat{g} \circ \hat{f}) = \inf_{v \in V : \hat{g}(v) = Y} \| v - \hat{f}(X) \|, \tag{6}$$

where $\hat{f}$ and $\hat{g}$ are approximate $f$ and $g$. This score measures the minimal discrepancy between any latent representation $v$ that could correctly predict $Y$ when processed by $\hat{g}$ and the actual latent representation $\hat{f}(X)$ produced by the model from input $X$.

It is usually complicated to calculate the score in Equation 6 due to the infimum operator. One solution is to directly apply the gradient descent mechanism in [40]: $u \leftarrow u - \eta \nabla (\hat{g}(u) - Y)^2$, however, this approach overlooks the characteristics of time series data and the adaptability of the update process. Therefore, we propose a weighted gradient descent mechanism

$$u \leftarrow u - \eta \tilde{w} \nabla (\hat{g}(u) - Y)^2, \tag{7}$$

where $\tilde{w}$ represents a vector of weights specifically designed to enhance the adaptability of the update process. These weights adjust the influence of each component of the gradient, enabling the latent vector $u$ to more effectively minimize the squared error between the predicted $\hat{g}(u)$ and actual targets $Y$. With learning rate $\eta$, these weighted adjustments prevent inefficient updates, fostering better convergence and ensuring well-defined non-conformity scores in the semantic feature space.

## 4.3 Adaptive Weight Adjustment

To ensure precise calibration of our model for time series data, it is imperative to meticulously define the weight terms $\tilde{w}$, thereby enabling the algorithm to concentrate on minimizing errors where they are most critical. We propose an adaptive weight adjustment scheme in the latent space, fundamentally inspired by the intuition that weights should be dynamically adjusted to prioritize semantic features more pertinent to the current prediction task. To effectively quantify this pertinence, we employ the principle of proximity in error terms, where smaller discrepancies between predicted and ground-truth values suggest higher relevance [5]. For time series data, the underlying processes often exhibit consistent behaviors or trends. Similar errors reflect this consistency, making them reliable indicators for future occurrences. As a result, we expect that similar errors should work best to predict the current error. Note that these error terms are measured in the semantic feature space, allowing us to leverage the rich representations of the base model. Our strategy to handle non-exchangeable conditions in time series is to replace the quantile of in standard CP with a weighted quantile:

$$Q_{1-\alpha} \left( \sum_{i=1}^{t} \tilde{w}_i \cdot \delta_{s_i} + \tilde{w}_{t+1} \cdot \delta_\infty \right). \tag{8}$$

Here, $\tilde{w}_i$ are weights assigned based on the similarity of their error terms to the current prediction error, $s_i$ are the non-conformity score calculated in the latent space, $t$ represents the size of calibration data, and $\delta_a$ denotes a point mass at $a$ which represents a discrete distribution:

$$P(X = x) = \begin{cases} 1 & \text{if } x = \alpha, \\ 0 & \text{if } x \neq \alpha. \end{cases} \tag{9}$$

In Eq. 8, we assign larger weights to more relevant data points which can significantly enhance the lower bound of empirical coverage (as shown in Equation 2). This insight guides our adaptation of the attention mechanism to assess and quantify the relevance of training data points to a

given test instance. The attention mechanism is designed to calculate weights based on the similarity of error terms, which are then used to adjust the influence of each data point in the final prediction model ([42]). The attention weights here are: $\tilde{w} \leftarrow$ AttentionWeights$(\hat{g}(u), Y)$, and AttentionWeights$(\hat{g}(u), Y)$ represents the attention weights calculated by the Transformer model, reflecting the similarity between the predicted output $\hat{g}(u)$ and $Y$. Accordingly, the attention mechanism in our approach actively assigns higher weights to these similar errors, thereby optimizing the inference coverage. The adaptive weight adjustment bolsters our method's capacity to consistently meet the significant level, providing a robust guarantee for its performance.

## 4.4 Weighted Conformal Prediction with Semantic Features

Since the non-conformity score is initially constructed in the semantic space, it is necessary to translate the prediction intervals to the output space for practical application. Our method incorporates band estimation techniques [40] to estimate confidence or prediction intervals around the forecasts. This approach allows for uniform uncertainty levels in the latent space, while effectively reflecting varied levels of uncertainty in the output space based on the non-linear transformation. In this work, we apply linear relaxation-based perturbation analysis (LiPRA) ([50]) to tackle this problem under deep neural network regimes. LiPRA transforms the certification problem as a linear programming problem, and solves it accordingly. Following [40], we model the Band Estimation problem as a perturbation analysis one, where we regard the surrogate feature as a perturbation of the trained feature and analyze the output bounds of the prediction head. Since LiPRA results in a relatively looser interval than the actual band, this method would give an upper bound estimation of the exact band length.

We summarize our framework in Algorithm 1. First, it calculates non-conformity scores within the latent space by splitting the base model into a feature function and a prediction head in step 3. By doing so, our method can utilize the latent features to generate precise and efficient prediction sets that are tailored to the specific dynamics of the data. During the calibration phase, the algorithm integrates the attention mechanism in steps 7-10 to refine the learning process of weights. This integration is crucial for dynamically prioritizing and adjusting the influence of specific features based on their relevance to the prediction task. Lastly, step 14 applies a structured process to achieve comprehensive coverage and maintain competitive intervals in our prediction sets. As a result, our method can dynamically adjust weights in response to changes in data patterns.

---

**Algorithm 1** Conformalized Time Series with Semantic Features

---

**Require:** Dataset $\{(X_t, Y_t)\}_{t=1}^T$, test feature $\{X_i\}_{i \in I_{te}}$, $\eta$, $M$, $\tilde{w}$, $\alpha$;
1: Randomly split the dataset $D$ into training $D_{tr} = \{(X_i, Y_i)\}_{i \in I_{tr}}$ and calibration $D_{ca} = \{(X_i, Y_i)\}_{i \in I_{ca}}$;
2: Training with $D_{tr}$
3: Train a base deep NN model for time series $\mu = \hat{g} \circ \hat{f}(\cdot)$ using the training fold $D_{tr}$;
4: Adaptive weight adjustment
5: $u \leftarrow \{\hat{f}(X_i)\}_{i \in I_{ca} \cup I_{te}}$; $m \leftarrow 0$; $n \leftarrow |I_{ca}| + |I_{te}|$; $\tilde{w} \leftarrow [1/n, \ldots, 1/n]$
6: **while** $m < M$ **do**
7:     predict $\hat{g}(u) - \{Y_i\}_{i \in I_{te}}$
8:     update $\tilde{w}$
9:     $u \leftarrow u - \eta\tilde{w}\nabla\left(\|\hat{g}(u) - \{Y_i\}_{i \in I_{ca} \cup I_{te}}\|^2\right)$
10:     $m \leftarrow m + 1$
11: **end while**
**Ensure:** $s(X, Y, \hat{g} \circ \hat{f}) = \|u - \hat{f}(X)\|$.
12: **for** $i \in I_{te}$ **do**
13:     Calibration with $D_{ca}$
14:     Calculate the $(1 - \alpha)$-th quantile $Q_{1-\alpha}$ of the distribution $\frac{1}{|D_{ca}|+1}\left(\sum_{i=1}\tilde{w}_i \cdot \delta_{s_i} + \tilde{w}_{n+1} \cdot \delta_\infty\right)$.
15:     Prediction
16:     Apply LiPRA on $\hat{f}(X_i)$ with perturbation $Q_{1-\alpha}$ and prediction head $\hat{g}$, which returns $C_{1-\alpha}(X_i)$;
17:     $D_{ca} \leftarrow X_i$
18: **end for**
**Ensure:** $C_{1-\alpha}(X_i)$ for each test input.

---

## 4.5 Theoretical Guarantee

This section outlines the theoretical guarantees for CT-SSF with respect to coverage and prediction interval length. Based on Theorem 4 in [40], we demonstrate that constructing non-conformity scores in the latent space allows CT-SSF to surpass CP for time series in the output space (e.g., NexCP) in terms of average prediction interval length under the following mild assumption:

**Assumption 1.** *In the output space, we define $H(v, X)$ as the length associated with sample $X$, derived from the length $v$ in the feature space. This is represented as $H(v, X) = \{g(u) \in \mathbb{R} : \|u - \hat{f}(X)\| \leq \frac{v}{2}\}$. We assume a Hölder condition for $H$, stipulating that for any $X$, the inequality $|H(v, X) - H(u, X)| \leq L|v - u|^{\beta}$ holds, where $\beta > 0$ and $L > 0$ are constants.*

The Hölder condition for $H(u, X)$ ensures that small changes in the input lengths $u$ or $v$ result in small and predictable changes in the length $H$. This is characterized by the constants $\alpha$ and $L$, which control the sensitivity of $H$ ([12]). In the context of CT-SSF, this condition helps maintain consistent lengths when transforming from the feature space to the output space (length preservation), ensures that differences between individual lengths and their quantiles are amplified (expansion), and guarantees that the prediction intervals remain stable and reliable across both spaces for a given calibration set (quantile stability). Below is an informal statement of the theoretical guarantee of CT-SSF. The complete version of the theorem is in Appendix C.1.

**Theorem 1.** *Under Assumption 1, if the following cubic conditions hold:*

1. ***Length Preservation***. *CT-SSF method does not cost much loss in feature space.*
2. ***Expansion***. *The Band Estimation operator expands the differences between individual length and their quantiles.*
3. ***Quantile Stability***. *The band length is stable in both the feature space and the output space for a given calibration set.*

*Let $\tilde{w}_i$ denote the weights learned from Algorithm 1. Suppose that $\tilde{w}_i$'s are independent of the calibration and test nonconformity scores. Then CT-SSF outperforms CP approaches for time series in the output space in terms of average band length while maintaining a valid coverage guarantee.*

Here, length preservation ensures that the transformation from feature space to output space maintains consistent interval lengths, minimizing efficiency loss. Expansion plays a key role in reducing inefficiency, as the latent space exhibits smaller distances between individual non-conformity scores and their quantiles, reducing the computational cost of the quantile operation. Lastly, quantile stability ensures that the interval is generalizable from the calibration set to the test samples. Since these conditions primarily emphasize the properties of quantiles and transformation steps rather than the reweighting, extending this framework to incorporate a weighted setting is a logical and justified progression. Note that in our theoretical results (for both efficiency and coverage) essentially assumes the weights to be fixed, where in implementation, the weights are learned from the data. It remains an interesting question to establish a condescending theory for data-drive weights. The detailed proofs can be found in Appendix C.1.

## 5 Experiments

### 5.1 Experimental Setup

**Prediction Models for Time Series**. CT-SSF is model agnostic, therefore, any NN-based prediction models for time series.prediction models can be used as the base models. To better show the advantage of our proposed method, we utilize a Recurrent Neural Network (RNN) model, which can be replaced with more advanced models like Transformers ([42]).

**Datasets.** Our experiments encompass evaluations on both synthetic and four real-world benchmark datasets, allowing for assessment under controlled and natural conditions. They are electricity, stock of Amazon, weather, and wind data. The details of these datasets can be found in Appendix B.

**Calibration Details.** During calibration, to get the best value for the number of steps $M$, we take a subset (e.g., one-fifth) of the calibration set as the additional validation set. We calculate the non-conformity score on the rest of the calibration set with various values of step $M$ and then evaluate

the validation set to get the best $M$ whose coverage is right above $1 - \alpha$. The final trained surrogate feature $v$ is close to the true feature because $\hat{g}(v)$ is sufficiently close to the ground truth $Y$. In practice, the surrogate feature after optimization satisfies

$$\frac{\|\hat{g}(v) - Y\|^2}{\|Y\|^2} < 1\%. \tag{10}$$

Eq. 10 indicates that the normalized error of predictions with the surrogate feature is less than 1%. Therefore, the surrogate feature approximates ground truth feature in the latent space.

**Compared Approaches.** Baselines include state-of-the-art (SOTA) methods for CP under distribution drift and SOTA CP for time series: (1) **Standard Split CP** ([43]). This is the standard split CP method outlined in Section 3; (2) **NexCP** ([7]). A refined CP method for handling distribution drifts using manually selected weights. Details are outlined in Section 3; (3) **FeatureCP (FCP)** ([40]). Implementing CP within a latent feature space under the assumption of exchangeability; (4) **HopCPT** ([5]). The current state-of-the-art CP for time series. HopCPT leverages Modern Hopfield Networks to build prediction intervals by identifying and utilizing historical events with similar error distributions.

**Metrics.** Our analysis employs two widely used metrics to assess the effectiveness of each CP method ([4]): *Empirical Coverage Rate (Coverage):* Measures the effectiveness of CP in achieving the theoretically guaranteed coverage. *Average Prediction Set Size (Width):* Evaluates the efficiency of CP, reflecting the compactness of the prediction intervals.

All the standard deviations are obtained over five repeated runs with different random seeds under the same base model. Code and data are available at `https://github.com/baiting0522/CT-SSF`.

## 5.2 Simulations

We adapt the simulation method in [7] and generate time series datasets that incorporate manually designed temporal dependency and heteroskedasticity. We generate $n = 1,000$ data points $(X_i, Y_i) \in \mathbb{R}^M \times \mathbb{R}^N$ by sampling $X_i$ from a Gaussian distribution, $X_i \sim \mathcal{N}(0, I_M)$. We set $Y_i \sim AX_i + BX_{i-1} + CX_{i-2} + \epsilon_i$. The coefficient matrix $A, B, C$ are set to the identity matrix $I_M$ and $\epsilon_i \sim \mathcal{N}(0, \Sigma)$ where $\Sigma$ is a diagonal matrix and its elements are given by $0.5 + 0.1i$.

| Methods | Miscoverage error | | | | | |
| | 0.05 | | 0.1 | | 0.15 | |
| | Cov | Width | Cov | Width | Cov | Width |
|---|---|---|---|---|---|---|
| CP | 94.80±1.83 | 10.39±0.49 | 90.90±1.50 | 9.30±0.11 | 84.90±2.83 | 8.24±0.40 |
| FCP | 94.20±0.68 | 1.77±0.27 | 89.70±2.41 | 1.72±0.21 | 85.10±1.32 | 1.40±0.21 |
| NexCP | 95.30±1.57 | 9.60±0.44 | 91.50±0.63 | 9.10±0.40 | 85.50±2.90 | 7.66±0.39 |
| HopCPT | 95.10±3.27 | 1.54±0.87 | 91.28±3.40 | 1.43±0.78 | 85.10±4.04 | 1.10±0.72 |
| CT-SSF | 96.50±0.84 | **1.45**±0.23 | 90.70±1.57 | **1.23**±0.15 | 85.70±2.14 | **1.04**±0.34 |

Table 1: Performance of the evaluated CP algorithms for the simulations dataset.

Table 1 presents the performance of the compared CP algorithms across three different levels of miscoverage error ($\alpha$): 0.05, 0.1, and 0.15. We observe from the results that CT-SSF consistently outperforms other baselines across all tested scenarios. Specifically, at a miscoverage level of 0.05, CT-SSF shows a 4.61% reduction in interval width compared to the SOTA approach, HopCPT. The smallest width indicates that our method can produce the most informative prediction intervals while maintaining valid empirical coverage. This efficiency is attributed to the innovative construction of the non-conformity score within the latent space with adaptive weight adjustment, which enhances the ability to leverage the underlying information of the base model. Meanwhile, the superior performance of FCP compared to CP and NexCP aligns with the experimental results reported in [40]. However, since FCP operates under the assumption of exchangeability, it may occasionally result in under-coverage, e.g., $\alpha = 0.05$ and $\alpha = 0.10$. On the other hand, NexCP tends to produce overly conservative results, though it maintains robust coverage. This underscores the need for our adaptive weight adjustments with semantic features to optimize performance.

**Ablation Study.** To further investigate the effectiveness of using semantic features and adaptive weight adjustment in CT-SSF, we compare it with its three variants that use: (1) Manually selected

weights in the Semantic space (CT-MS), (2) Manually selected weights in the Output space (CT-MO). This variant reduces to the NexCP method and (3) Learned weights in the Output space (CT-LO). The weights are learned using the same learning scheme in CT-SSF.

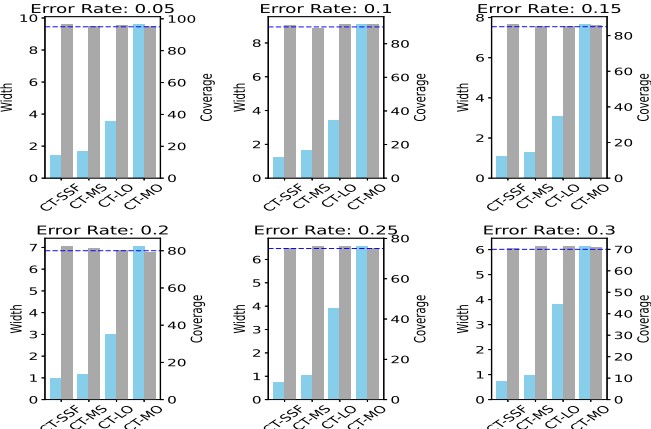

Figure 1: Comparisons of the variants of CT-SSF. The blue bar chart represents Width and the gray bar represents Coverage, the blue line represents target coverage.

Figure 1 demonstrates the effectiveness of the CT-SSF method compared to its variants across varied miscoverage risks from 0.05 to 0.3. CT-SSF consistently outperforms the other methods in terms of prediction interval width, indicating its superior efficiency in generating compact prediction sets while maintaining robust empirical coverage. Relative to CT-MO and CT-LO, our method achieves more than a 50% reduction in the width of the prediction intervals, a benefit derived from exploiting the rich latent information in the feature space. Additionally, in comparison with CT-MS, our method shows an approximate 10% reduction in interval width. These results demonstrate the significant impact of adaptive weight adjustments on the efficiency of prediction intervals, further establishing CT-SSF as a highly effective CP method for time series forecasting.

CT-SSF's performance can be attributed to its specialized approach to learning weights in the latent space. Unlike methods such as NexCP and CT-MS, which apply predefined weight schemes, CT-SSF's dynamic adaptation to the underlying data structure allows for more precise adaptation of the model to the specific data characteristics. Unlike HopCPT, which learns the weights but constructs non-conformity scores in the output space, CT-SSF builds these scores directly within the latent space to utilize underlying latent features effectively and better capture the data variability and structural dependencies, resulting in more reliable and informative prediction intervals.

## 5.3 Real-world Data

Real-world experiments are conducted using benchmark time series data including *electricity* ([7]), *stock market* ([14, 2]), *weather* ([28, 54]), and *wind speed forecasting* ([48]).

|         |       | CT-SSF | HopCPT | NexCP | FCP | CP |
|---------|-------|--------|--------|-------|-----|-----|
| Elec    | Cov   | 90.59±1.74 | 90.88±2.39 | 91.67±0.79 | 91.26±0.99 | 91.26±1.15 |
|         | Width | **0.21**±0.03 | 0.23±0.05 | 0.74±0.17 | 0.28±0.05 | 0.83±0.01 |
| Stock   | Cov   | 91.58±2.23 | 90.79±3.06 | 91.03±±2.37 | 91.58±2.07 | 91.74±1.98 |
|         | Width | **0.19**±0.03 | 0.23±0.04 | 1.43±0.12 | 0.29±0.10 | 1.51±0.17 |
| Weather | Cov   | 90.12±0.26 | 90.11±0.68 | 90.14±0.26 | 90.06±0.22 | 90.13±0.26 |
|         | Width | **0.012**±0.002 | 0.017±0.004 | 0.034±0.006 | 0.023±0.004 | 0.034±0.007 |
| Wind    | Cov   | 90.32±1.51 | 90.06±2.49 | 90.06±2.08 | 90.19±1.75 | 89.54±1.75 |
|         | width | **0.52**±0.03 | 0.54±0.04 | 2.76±0.11 | 0.59±0.13 | 2.65±0.14 |

Table 2: Performance of the evaluated CP algorithms for the real data. The specified miscoverage level is $\alpha = 0.1$ for all experiments. Results for different $\alpha$ can be found in Appendix B.

Results in Table 2 reveal that CT-SSF maintains valid coverage rates while excelling in minimizing prediction interval widths. Across all datasets, CT-SSF shows a 10%-20% reduction in prediction intervals compared to the SOTA HopCPT. While NexCP is designed for non-exchangeable settings, the need for manually selected weights limits its practical application. For instance, in the *wind* dataset, improperly chosen weights result in larger prediction intervals compared to standard CP, despite meeting theoretical coverage guarantees. We further conduct an ablation study to compare the variants of CT-SSF, similar to that for synthetic data. Results in Appendix B show that CT-SSF outperforms other variants. This demonstrates the effectiveness of utilizing semantic features and implementing adaptive weight adjustments. While our experiments are based on RNN, we show in Appendix B that using a different base model can lead to similar findings.

|         |       | 2nd | 3rd | 4th | 5th | 6th |
|---------|-------|-----|-----|-----|-----|-----|
| Elec    | Cov   | 89.75±1.96 | 90.94±1.02 | 90.59±1.74 | 90.22±2.02 | 90.21±1.92 |
|         | Width | 0.23±0.09 | 0.29±0.02 | **0.21**±0.03 | 0.28±0.02 | 0.30±0.02 |
| Stock   | Cov   | 93.50±1.52 | 88.65±2.45 | 91.58±2.23 | 88.50±2.42 | 88.41±2.60 |
|         | Width | 0.39±0.12 | 0.26±0.09 | **0.19**±0.03 | 0.21±0.07 | 0.21±0.07 |
| Weather | Cov   | 90.12±0.31 | 90.07±0.34 | 90.12±0.26 | 90.06±0.56 | 90.06±0.54 |
|         | Width | 0.018±0.004 | 0.034±0.010 | **0.012**±0.002 | 0.014±0.002 | 0.014±0.002 |
| Wind    | Cov   | 89.67±2.42 | 90.32±1.51 | 89.28±1.88 | 88.63±1.90 | 88.90±1.80 |
|         | Width | 1.00±.023 | **0.52**±0.03 | 0.54±0.12 | 0.59±0.03 | 0.65±0.02 |

Table 3: Performance of the CT-SSF with different $f$ and $g$ configurations. The specified miscoverage level is $\alpha = 0.1$ for all experiments.

**The impact of choosing $f$ and $g$.** Finally, we analyze the potential impact of the selection of the semantic feature space on the performance of CT-SSF. Our base model is an 8-layer RNN, and we conduct experiments on semantic spaces ranging from the 2nd to the 6th layer, i.e., using the first 2-6 layers as $f$ and the rest as $g$. Results in Table 3 indicate that the performance of CT-SSF is influenced by the configuration of $f$ and $g$, as evidenced by the variation in the length of prediction intervals. A key observation is that optimal performance typically arises from the middle layers (specifically, the 3rd and 4th layers). This suggests that $f$ with too few layers may not capture enough semantic information, and $g$ with a limited number of layers lacks adequate predictive strength. Additionally, the coverage and interval length results are similar for the 5th and 6th layers across all four datasets. This similarity may be due to the comparable level of information interpreted in these feature spaces. Given all these observations, it is important to perform cross-validation to identify the best semantic space for the highest efficiency when deployed in various applications.

# 6 Conclusions and Limitations

In this work, we introduce CT-SSF, a novel approach designed to overcome the challenges of applying CP directly to time-series data. Our method shifts the calculation of non-conformity scores to the semantic feature space, effectively capturing the complex, nonlinear relationships in time-series data that are often overlooked in the output space. By employing a reweighting scheme and dynamically adjusting weights based on the importance of various semantic features, CT-SSF achieves valid coverage with high efficiency. We provide theoretical analyses demonstrating that CT-SSF outperforms CP methods that operate in the output space. Experimental results show that CT-SSF significantly outperforms existing SOTA CP methods for time-series data.

Our approach to CP for time series forecasting presents certain limitations that warrant further exploration and mitigation. Firstly, its reliance on NNs excludes simpler models like ridge regression or random forests. This limitation, while justified by the prevalent use of NN in time series forecasting, narrows the scope of our methodology's wide applicability. Secondly, the robustness of our prediction set lengths is sensitive to the configuration of the latent space to construct $f$ and $g$. Our current solution uses a validation set to help separate $f$ and $g$, rendering less data for calibration. Future enhancements could include the development of adaptive methods that dynamically refine the latent space to enhance the stability of the predictions across diverse data scenarios, thereby addressing the current limitations and expanding the utility of our approach.

## Acknowledgements

This work is supported by the National Science Foundation (NSF) Grant #2312862, National Institutes of Health (NIH) #R01AG091762, and a Cisco gift grant.

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

# A    Comparisons to existing methods

This section contains descriptions of the relationship of our method to existing approaches of conformal prediction for time series data. There are generally three primary approaches designed to manage these challenges and enhance the reliability and validity of the CP in time series: reweighting, updating non-conformity scores and updating miscoverage level. We introduce representative work for each of these categories here. For a more detailed analysis of CP methods for time series, please refer to [57].

Reweighting assigns relevance-based weights to data points to align the data distribution closer to a target distribution. NexCP [7] exponentially decayed weights to emphasize recent observations, but these lack adaptability. HopCPT [5] improves on this by using a Modern Hopfield Network (MHN) for similarity-based reweighting. It assigns weights to past time steps based on their relevance to the current time step. Encoded inputs are processed with learned weight matrices, and a hyperparameter adjusts the focus of the probabilistic distribution. These weights create weighted conformal prediction intervals by discounting extremal quantiles of relative errors.

The second technique, updating non-conformity scores leverages the most recent $T$ data points and continuously updates prediction intervals as new data becomes available. For example, EnbPI [48] updates the non-conformity score with sequential error patterns to adapt the intervals dynamically. And SPCI [49] replaces the empirical quantile with an estimate by a conditional quantile estimator to effectively address serial dependencies among residuals in sequential analysis.

The last main direction for CP in time series focuses on adaptively adjusting the significance level $\alpha$ during test time to account for mis-coverage. This method can dynamically adjust the size of prediction sets in an online setting where the data generating distribution is allowed to vary over time in an unknown fashion. For example, the update rule for the quantile level $\alpha$ in ACI [14] is:

$$\alpha_{t+1} = \alpha_t + \gamma(\alpha - \text{err}_t), \tag{11}$$

where $\gamma$ is a step size parameter, and $\text{err}_t$ indicates if $Y_t$ was not included in $\hat{C}_t(\alpha_t)$. The approach ensures that the prediction intervals adjust over time to account for shifts in the data distribution, maintaining the desired coverage probability.

Our work falls under a reweighting scheme, with a focus on weighted conformal prediction in the latent feature space. This approach enables us to capture the underlying data structure more precisely, leading to tighter prediction intervals. We leverage the standard attention mechanism for its simplicity and computational efficiency, while also allowing for dynamic adjustment of data point significance. Furthermore, our method effectively integrates attention-based weight updates within the latent feature space, where the learned rich representations capture the intrinsic characteristics of the data with greater accuracy.

# B    Extended Experiments

**Datasets Details**

Datasets from four different domains are used in the experiments. Details are summarized below.

- **Electricity.** The Electricity Demand Forecasting dataset, introduced by Harries in 1999 ([18]), tracks electricity usage and pricing in New South Wales and Victoria, Australia, with half-hourly data from May 7, 1996, to December 5, 1998. For our experiment, we focus on four key variables: nswprice and vicprice (electricity prices in New South Wales and Victoria) and nswdemand and vicdemand (electricity demand in each state). Our response variable, transfer, measures the electricity transferred between the two states. We select data from 9:00 AM to 12:00 PM to reduce daily fluctuations and discard an initial period with constant transfer values, resulting in 3,444 time points for analysis.

- **Amazon Stock.** The Amazon stock dataset provides detailed historical data on Amazon's stock performance. This dataset includes key financial metrics such as the opening price, highest price, lowest price, closing price, and trading volume for each trading day. The dataset captures daily stock prices at various intervals, providing a comprehensive view of Amazon's stock market behavior. These metrics are crucial for analyzing trends, volatility,

and the overall performance of Amazon's stock over time. The data is structured with columns representing the opening price (open), the highest price during the day (high), the lowest price during the day (low), the closing price (close), and the volume of shares traded (volume).

- **Weather.** The weather dataset records meteorological data every 10 minutes throughout the entire year of 2020, encompassing 21 key indicators that provide a comprehensive overview of weather conditions. Key indicators include air temperature, humidity, wind direction, rainfall, solar radiation. This dataset offers detailed, granular insights into weather patterns and conditions, making it valuable for meteorological analysis and forecasting.

- **Wind Speed.** The wind speed data are collected at wind farms operated by the Midcontinent Independent System Operator (MISO) in the US ([59]). The dataset records wind speed measurements updated every 15 minutes over a one-week period in September 2020. This high-frequency data provides detailed insights into wind speed variations and is crucial for analyzing and modeling wind energy production.

**Additional Experiments** We conduct extended experiments on the real-world data for RNN and a different base model for time series prediction, Transformer. Results with different miscoverage levels can be seen in Table 4 and Table 5, respectively. We also perform an ablation study similar to Section 5.2 for real data in Figure 2. CT-SSF consistently delivers the shortest prediction intervals across all datasets and varying error rates when utilizing a Transformer as the base model. Although it does not perform optimally for *weather* and *wind* data at a 0.05 error rate with an RNN base, CT-SSF still achieves results comparable to the top-performing method, HopCPT. These experiments support the conclusion that CT-SSF can produce competitive prediction intervals while maintaining coverage guarantee.

| | error rate | | CT-SSF | HopCPT | NexCP | FCP | CP |
|---|---|---|---|---|---|---|---|
| **Elec** | 0.05 | Cov | 95.32±0.02 | 95.18±1.40 | 95.27±0.54 | 95.52±1.04 | 95.47±1.09 |
| | | Width | **0.23**±0.04 | 0.26±0.07 | 0.98±0.04 | 0.34±0.06 | 0.99±0.02 |
| | 0.15 | Cov | 84.87±1.81 | 84.90±1.75 | 85.57±0.70 | 86.67±1.48 | 86.27±1.47 |
| | | Width | **0.20**±0.03 | 0.23±0.05 | 0.62±0.04 | 0.25±0.04 | 0.73±0.01 |
| | 0.20 | Cov | 80.12±2.06 | 79.79±1.88 | 80.93±0.70 | 81.04±1.77 | 81.16±1.97 |
| | | Width | **0.19**±0.03 | 0.20±0.04 | 0.60±0.03 | 0.22±0.04 | 0.65±0.01 |
| **Stock** | 0.05 | Cov | 95.47±1.51 | 95.32±2.21 | 95.23±1.27 | 95.31±1.33 | 95.23±1.35 |
| | | Width | **0.30**±0.04 | 0.31±0.05 | 2.13±0.28 | 0.45±0.17 | 2.27±0.22 |
| | 0.15 | Cov | 86.75±2.78 | 85.71±2.09 | 86.03±2.09 | 86.90±2.40 | 86.42±2.53 |
| | | Width | **0.18**±0.03 | 0.20±0.04 | 1.18±0.12 | 0.24±0.07 | 1.21±.06 |
| | 0.20 | Cov | 80.87±1.90 | 81.74±3.50 | 81.03±3.56 | 82.93±2.02 | 82.61±1.38 |
| | | Width | **0.15**±0.02 | 0.18±0.04 | 0.87±0.09 | 0.20±0.06 | 1.07±0.02 |
| **Weather** | 0.05 | Cov | 95.08±0.20 | 94.96±0.66 | 95.14±0.20 | 95.01±0.17 | 94.97±0.17 |
| | | Width | 0.018±0.002 | **0.017**±0.002 | 0.038±0.006 | 0.023±0.004 | 0.044±0.009 |
| | 0.15 | Cov | 85.24±0.37 | 85.29±0.79 | 85.37±0.54 | 84.90±0.22 | 84.90±0.16 |
| | | Width | **0.012**±0.002 | 0.014±0.002 | 0.032±0.005 | 0.015±0.003 | 0.029±0.006 |
| | 0.20 | Cov | 80.31±0.26 | 80.38±0.78 | 80.20±0.50 | 79.89±0.21 | 79.89±0.28 |
| | | Width | **0.011**±0.002 | 0.013±0.002 | 0.031±0.006 | 0.013±0.003 | 0.025±0.005 |
| **Wind** | 0.05 | Cov | 95.03±1.58 | 96.07±1.24 | 95.94±1.62 | 93.85±1.73 | 94.38±1.68 |
| | | Width | 0.67±0.06 | **0.66**±0.10 | 3.28±0.24 | 0.69±0.14 | 3.15±0.28 |
| | 0.15 | Cov | 84.18±2.90 | 85.36±2.63 | 85.75±3.07 | 84.96±1.43 | 83.13±2.59 |
| | | Width | **0.42**±0.05 | 0.45±0.03 | 2.25±0.17 | 0.52±0.12 | 2.28±0.07 |
| | 0.20 | Cov | 77.38±3.32 | 78.56±6.16 | 79.60±4.50 | 78.82±1.87 | 76.20±3.98 |
| | | Width | **0.37**±0.05 | 0.40±0.06 | 1.98±0.09 | 0.46±0.10 | 2.05±0.12 |

Table 4: Performance of the evaluated CP algorithms for the real data of RNN. The standard deviation is obtained over five repeated runs with different random seeds.

## C  Theoretical Guarantee

This section provides theoretical guarantees for CT-SSF regarding coverage and interval length (i.e., efficiency). We use the seminar work NexCP for illustration and the results below can be applied to other CP for time series in the output space as well.

### C.1  Efficiency

**Notations.** Let $P$ denote the population distribution. Let $D_{ca} \sim \mathbb{P}_n$ denote the calibration set with sample size $n$, where we overload the notation $P_n$ to denote the distribution of a set with samples drawn from a distribution $P$. Given the model $g \circ f$ with feature extractor $f$ and prediction head $g$, we assume $g$ is continuous. We also overload the notation $Q_{1-\alpha}(V)$ to denote the $(1-\alpha)$-quantile of the set $V \cup \{\infty\}$. Here, the set $V$ represents the empirical distribution after reweighting, i.e., $\sum_{i=1}^{t} \tilde{w}_i \cdot \delta_{v_i} + \tilde{w}_{t+1} \cdot \delta_{\infty}$. Additionally, let $\mathbb{M}[\cdot]$ denote the mean of a set, and let a set minus a real number denote the broadcast operation.

**NexCP.** Let $V_{D_{ca}}^o = \{v_i^o\}_{i \in I_{ca}}$ denote the individual lengths in the output space, given the calibration set $D_{ca}$. Specifically, $v_i^o = 2|y_i - \hat{y}_i|$, where $y_i$ denotes the true response of sample $i$ and $\hat{y}_i$ denotes the corresponding prediction. Since NexCP returns band length with the $1 - \alpha$ quantile of the non-conformity score, the resulting average band length is derived as $Q_{1-\alpha}(V_{D_{ca}}^o)$.

| | error rate | | CT-SSF | HopCPT | NexCP | FCP | CP |
|---|---|---|---|---|---|---|---|
| **Elec** | 0.05 | Cov | 95.42±1.43 | 95.50±1.43 | 95.58±0.42 | 95.58±1.06 | 95.47±1.09 |
| | | Width | **0.19**±0.06 | 0.26±0.07 | 0.78±0.03 | 0.28±0.05 | 0.98±0.02 |
| | 0.10 | Cov | 90.59±1.74 | 90.88±2.39 | 90.22±2.02 | 91.23±0.86 | 91.26±1.00 |
| | | Width | **0.17**±0.05 | 0.23±0.05 | 0.62±0.04 | 0.24±0.04 | 0.84±0.01 |
| | 0.15 | Cov | 85.05±1.65 | 85.74±1.23 | 86.30±1.56 | 84.96±1.77 | 86.27±1.47 |
| | | Width | **0.15**±0.04 | 0.21±0.04 | 0.58±.03 | 0.21±0.02 | 0.73±0.03 |
| **Stock** | 0.05 | Cov | 95.16±1.61 | 95.24±1.28 | 95.31±1.21 | 95.07±1.43 | 95.23±1.35 |
| | | Width | **0.21**±0.04 | 0.30±0.05 | 1.83±0.05 | 0.35±0.11 | 2.27±0.21 |
| | 0.10 | Cov | 91.67±2.07 | 90.79±2.09 | 91.27±1.91 | 91.42±2.14 | 91.74±1.98 |
| | | Width | **0.16**±0.04 | 0.23±0.04 | 1.23±0.16 | 0.24±0.08 | 1.51±0.17 |
| | 0.15 | Cov | 85.71±2.14 | 86.74±2.78 | 86.03±2.09 | 96.50±2.37 | 86.43±2.53 |
| | | Width | **0.14**±0.03 | 0.20±0.04 | 0.89±0.05 | 0.19±0.05 | 1.21±0.06 |
| **Weather** | 0.05 | Cov | 94.97±0.17 | 95.04±0.24 | 95.08±0.24 | 94.96±0.16 | 94.97±0.17 |
| | | width | **0.019**±0.004 | 0.020±0.006 | 0.038±0.004 | 0.023±0.003 | 0.044±0.009 |
| | 0.10 | Cov | 90.11±0.19 | 90.20±0.50 | 90.23±0.37 | 90.19±3.72 | 90.13±0.26 |
| | | Width | **0.015**±.003 | 0.017±0.005 | 0.032±0.006 | 0.021±0.003 | 0.035±0.007 |
| | 0.15 | Cov | 84.89±0.21 | 85.29±0.48 | 85.37±0.55 | 85.24±0.37 | 84.90±0.16 |
| | | Width | **0.012**±0.003 | 0.014±0.005 | 0.027±0.004 | 0.015±0.003 | 0.029±0.006 |
| **Wind** | 0.05 | Cov | 95.81±1.78 | 95.03±1.58 | 95.81±1.63 | 94.25±2.00 | 94.38±1.68 |
| | | Width | **0.47**±0.04 | 0.67±0.05 | 3.76±0.15 | 0.55±0.12 | 3.15±0.28 |
| | 0.10 | Cov | 89.80±2.50 | 90.06±1.72 | 90.12±1.23 | 90.33±1.51 | 89.54±1.75 |
| | | Width | **0.37**±0.04 | 0.52±0.04 | 2.91±0.11 | 0.46±0.10 | 2.65±0.14 |
| | 0.15 | Cov | 84.83±2.78 | 84.18±2.90 | 85.62±2.89 | 83.14±2.59 | 84.83±1.20 |
| | | Width | **0.39**±0.03 | 0.42±0.04 | 2.68±0.37 | 0.41±0.09 | 2.28±0.07 |

Table 5: Performance of the evaluated CP algorithms for the real data with Transformer as the base model. The standard deviation is obtained over five repeated runs with different random seeds.

**CT-SSF.** Let $V_{D_{ca}}^f = \{v_i^f\}_{i \in I_{ca}}$ be the individual lengths in the semantic feature space given the calibration set $D_{ca}$. The resulting band length in CT-SSF is denoted by $\mathbb{E}_{X_{test}, Y_{test}} \left[ H\left( Q_{1-\alpha}(V_{D_{ca}}^f), X_{test} \right) \right]$.

We propose a formal description of the cubic conditions and provide the proof following the structure of [40].

**Theorem 1.** Assume Assumption 1 holds. Additionally, we assume that there exist constants $\epsilon > 0$ and $c > 0$, such that the feature space satisfies the following cubic conditions:

1. **Length Preservation.** Semantic-Feature CP does not incur a significant loss in the feature space in a quantile manner:

$$\mathbb{E}_{D \sim P_n} Q_{1-\alpha}(H(V_D^f, D)) < \mathbb{E}_{D \sim P_n} Q_{1-\alpha}(V_D^o) + \epsilon.$$

2. **Expansion.** The operator $H(v, X)$ expands the differences between individual lengths and their quantiles, namely,

$$L\mathbb{E}_{D \sim P_n} \left[ \mathbb{M} \left[ Q_{1-\alpha}(V_D^f) - V_D^f \right]^\alpha \right] < \mathbb{E}_{D \sim P_n} \left[ \mathbb{M} \left[ Q_{1-\alpha}(H(V_D^f)) - H(V_D^f) \right] \right]$$
$$- \epsilon - 2\max\{L, 1\} \left( \frac{c}{\sqrt{n}} \right)^{\min\{\alpha, 1\}}.$$

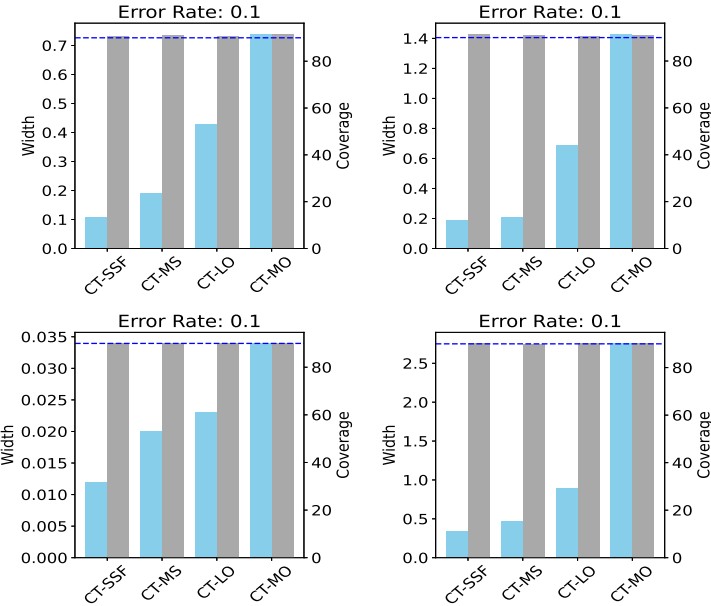

Figure 2: Comparisons of the variants of CT-SSF using the real datasets. The specified miscoverage level is $\alpha = 0.1$. The blue bar chart represents Width and the gray bar represents Coverage, the blue line represents the target coverage.

3. **Quantile Stability.** Given a calibration set $D_{ca}$, the quantile of the band length is stable in both the feature space and output space, namely,

$$\mathbb{E}_{D \sim P_n} \left[ Q_{1-\alpha}(V_D^f) - Q_{1-\alpha}(V_{D_{ca}}^f) \right] \leq \frac{c}{\sqrt{n}}$$

and

$$\mathbb{E}_{D \sim P_n} \left[ Q_{1-\alpha}(V_D^o) - Q_{1-\alpha}(V_{D_{ca}}^o) \right] \leq \frac{c}{\sqrt{n}}.$$

Let $\tilde{w}_i$ denote the weights learned from Algorithm 1. Suppose that $\tilde{w}_i$'s are independent of the calibration and test nonconformity scores. CT-SSF provably outperforms NexCP in terms of average band length, namely,

$$\mathbb{E} \left[ H(Q_{1-\alpha}(V_{D_{ca}}^f), X_{test}) \right] < Q_{1-\alpha}(V_{D_{ca}}^o),$$

where the expectation is taken over the calibration fold and the testing point $(X_{test}, Y_{test})$.

Before proceeding with the proof, we note that Theorem 1 extends Theorem 4 from [40] to accommodate non-exchangeable settings. This adaptation is feasible because the foundational assumptions and methodologies hinge primarily on quantile calculations rather than the specific characteristics of the underlying distributions. Thus, we can leverage the framework of their proof to construct our argument effectively.

*Proof.* By the expansion assumption,

$$L \mathbb{E}_D \mathbb{M} \left[ Q_{1-\alpha}(V_D^f) - V_D^f \right]^\alpha < \mathbb{E}_D \mathbb{M} \left[ Q_{1-\alpha}(H(V_D^f, D)) - H(V_D^f, D) \right] - \epsilon - 2 \max\{L, 1\} \left( \frac{c}{\sqrt{n}} \right)^{\min\{\alpha, 1\}}.$$

We rewrite it as

$$\mathbb{E}_D \mathbb{M} H(V_D^f, D) < \mathbb{E}_D Q_{1-\alpha}(H(V_D^f, D)) - \epsilon - 2 \max\{L, 1\} \left( \frac{c}{\sqrt{n}} \right)^{\min\{\alpha, 1\}} - L \mathbb{E}_D \mathbb{M} \left[ Q_{1-\alpha}(V_D^f) - V_D^f \right]^\alpha.$$

Due to the Hölder condition, we have that

$$\mathbb{M} H(Q_{1-\alpha}(V_D^f), D) < \mathbb{M}(H(V_D^f, D)) + L \mathbb{M} \left| Q_{1-\alpha}(V_D^f) - V_D^f \right|^\alpha,$$

therefore

$$\mathbb{E}_D \mathbb{M} \left[ H(Q_{1-\alpha}(V_D^f), D) \right] < \mathbb{E}_D Q_{1-\alpha} \left( H(V_D^f, D) \right) - \epsilon - 2\max\{1, L\} \left( \frac{c}{\sqrt{n}} \right)^{\min\{1,\alpha\}}.$$

Therefore, due to Assumption 1, we have that

$$\mathbb{E}_D \mathbb{M} H(Q_{1-\alpha}(V_D^f), D) < \mathbb{E}_D Q_{1-\alpha}(V_D^o) - 2\max\{1, L\} \left( \frac{c}{\sqrt{n}} \right)^{\min\{1,\alpha\}}.$$

Additionally, according to the quantile stability assumption, we have that

$$\mathbb{E}_D \mathbb{M} \left| H(Q_{1-\alpha}(V_D^f), D) - H(Q_{1-\alpha}(V_{D_{ca}}^f), D) \right| \le L \left( \frac{c}{\sqrt{n}} \right)^{\alpha}$$

and

$$\mathbb{E}_D \left[ Q_{1-\alpha}(V_D^o) - Q_{1-\alpha}(V_{D_{ca}}^o) \right] \le \frac{c}{\sqrt{n}}.$$

Therefore,

$$\mathbb{E} H(Q_{1-\alpha}(V_{D_{ca}}^f), X') = \mathbb{E}_D \mathbb{M} H(Q_{1-\alpha}(V_{D_{ca}}^f), D)$$

$$< Q_{1-\alpha}(V_{D_{ca}}^o) - 2\max\{1, L\} \left( \frac{c}{\sqrt{n}} \right)^{\min\{1,\alpha\}} + L \left( \frac{c}{\sqrt{n}} \right)^{\alpha} + \frac{c}{\sqrt{n}}$$

$$< Q_{1-\alpha}(V_{D_{ca}}^o).$$

$\square$

## C.2 Coverage Guarantee

Since the coverage for CP depends on the ranking of the nonconformity measures rather than their actual values ([35, 37]), we can extend the work of [7] to the semantic feature space, where we construct the nonconformity score and compute the corresponding quantile value.

**Theorem 2** (Coverage Gap; Adaptation from [7], Theorem 2). *Let $\Delta Cov$ denote the coverage gap between the empirical coverage and the desired coverage level. Let $\tilde{w}_i$ denote the weights learned from Algorithm 1. Suppose that $\tilde{w}_i$'s are independent of the calibration and test nonconformity scores. Then,*

$$\Delta Cov \ge - \sum_{i=1}^{t} \tilde{w}_i \cdot d_{TV}(R(Z), R(Z^i)),$$

*where $d_{TV}$ represents the total variation distance, and $R(Z)$ denotes the sequence of absolute residuals on $Z$ and $Z^i$ denotes a sequence where the test point $Z_{t+1}$ is swapped with the $i$-th calibration point. Note that these sequences are represented by the semantic features here.*

This coverage gap suggests that assigning lower weights to time steps with high total variation distance can help maintain small coverage gaps, reinforcing the theoretical foundation of our method. CT-SSF employs an attention mechanism to allocate higher weights to data points from the same error distribution as the current one and lower weights otherwise. According to Theorem 2, our dynamic weighting strategy can effectively minimize the coverage gap.

## D  Potential Social Impact

Our method significantly can enhance decision-making in finance, healthcare, and environmental monitoring by improving the accuracy and reliability of time series predictions. The success of our method, however, depends on the quality of training data used. This highlights the critical need for robust data management to maintain data integrity in dynamic environments. By upholding high data quality standards, our method ensures reliable support across these vital sectors, fostering trust in automated decision-making systems.

