# OpenReview forum: "Conformalized Time Series with Semantic Features"
_NeurIPS.cc/2024/Conference — NeurIPS 2024 poster_

### Official Review · Reviewer_8Q7C · 2024-06-23

**Soundness:** 2
**Presentation:** 2
**Contribution:** 2
**Rating:** 4
**Confidence:** 4

**Summary:**

The paper proposes producing conformal intervals using non-conformity scores defined in latent space, demonstrating improved performance against existing CP methods that work in output space.

**Strengths:**

Adopting CP in feature space rather than latent space and demonstrating performance gains is novel.

**Weaknesses:**

1. Model-agnostic: the authors claim CT-SSF is also model agnostic, yet the experimental results in Table 3 clearly show variation in terms of different f and g configurations. There arises the question on the wide applicability and computational efficiency of the method, especially if searching for the "optimal" (f,g) is costly.

2. Experimental comparison: how are the baselines implemented? Are the same RNN model used as the base model for the baselines? It is important to highlight what are kept the same across baselines in order to explain the performance gain of CT-SSF. If multiple components are varying, it is hard to attribute the performance gain to the use of latent features in CT-SSF.

3. Theory:

    - Section 4.5 tries to explain the theoretical guarantee of the proposed method, yet it is too hasty to be clear. The key question of why interval widths can be reduced lacks an intuitive explanation, and given the experimental results, it also seems such reduction depends on (f,g) combination, which is not clearly explained in the theory section.

   - What are assumptions on the original series? Does the guarantees always hold for any time series?

Instead of presenting Theorem 1 in this casual form, it is better to give an informal theoretical statements. The exact bound can be described in big-O notation.

**Questions:**

No more questions

**Limitations:**

See weaknesses above.

---

> ### Author Rebuttal · Authors · 2024-08-06
>
> We appreciate your valuable and informative suggestions and would clarify the questions and concerns in the following.
>
> Q1: Model-agnostic and f,g splitting criterion.
>
> A1: CT-SSF is model agnostic because CP is naturally model agnostic and any NN-based time series prediction models can be used as the base models.
>
> One splitting criterion of $f$ and $g$ to significantly reduce computational efficiency is using cubic metrics of different splitting points for selection. As shown in Figure 1 in the attached PDF, we plot the relationship between prediction efficiency (band length) and the cubic metric. The cubic metric here represents the core statement in the cubic condition (statement 2), which implies a metric form like $\mathbb{M}[Q_{1-\alpha}V^f_{D_{ca}}- V^f_{D_{ca}}]$. We find that the cubic condition is closely related to the performance of CT-SSF. For each given dataset, the cubic metric is negatively related to the band length and thus positively related to prediction efficiency. Given this positive relationship, it is possible to use the cubic metric as the criterion for cross-validation instead of directly optimizing for prediction efficiency. By focusing on the cubic metric during cross-validation, we can reduce the number of necessary computations and improve the overall efficiency of the model selection process.
>
> Additionally, the splitting criterion of $f$ and $g$ can follow standard cross-validation to find the best splitting point, like standard hyperparameter search. The process can be conducted on a smaller subset of the data to quickly estimate the best splitting point, thereby reducing the overall computational burden. Despite the variation, the experimental results on different splitting points consistently outperform other CP for time series benchmarks.
>
> Q2: Experimental Comparison.
>
> A2: A consistent RNN model is used as the base model across all baselines. After training the RNN model, it is fixed to ensure a fair comparison between different CP methods. This consistency allows us to attribute the performance gains specifically to the use of latent features in CT-SSF. The results demonstrate the excellence of CT-SSF in achieving superior prediction interval performance. To see if CT-SSF's improvement is consistent across different base models, we also conducted experiments on Transformers (see Table 5, Appendix A.1) and new experiments on CNN (see Table 1 in the attached PDF). Results for both architectures suggest that CT-SSF can outperform HopCPT with 10\% to 20\% shorter prediction intervals across all datasets.
>
> Q3: Theory.
>
> A3: A formal and detailed discussion regarding the theorem can be found in Appendix A.2. Here, we briefly summarize it.
>
> Theorem 1. Assume Assumption 1 holds. Additionally, we assume that there exist constants $ \epsilon > 0 $ and $ c > 0 $, such that the feature space satisfies the following cubic conditions:
> 1. Length Preservation. Semantic-Feature CP does not incur a significant loss in the feature space in a quantile manner, namely,
> $   \\mathbb{E} Q_{1-\alpha}(H(V^f_D, D)) \geq \mathbb{E} Q_{1-\alpha}(V^o_D) + \epsilon.$
> 2. Expansion. The operator $ H(v, X) $ expands the differences between individual lengths and their quantiles, namely,
> $L\\mathbb{E} [ \\mathbb{M} [ Q_{1-\alpha}(V^f_D)- (V^f_D)]^\alpha]<\mathbb{E}[ \mathbb{M} [ Q_{1-\alpha}(H(V^f_D)) - H(V^f_D)]- \epsilon - 2 \max \\{L, 1\\}( \frac{c}{\sqrt{n}})^{\min\\{\alpha,1\\}}.$
> 3. Quantile Stability. Given a calibration set $D^{'}$, the quantile of the band length is stable in both the feature space and output space, namely,$\mathbb{E} [ Q_{1-\alpha}(V^f_D) - Q_{1-\alpha}(V^f_{D^{'}} )]\leq \frac{c}{\sqrt{n}} $ and $\mathbb{E} [ Q_{1-\alpha}(V^o_D) - Q_{1-\alpha}(V^o_{D^{'}} )]\leq \frac{c}{\sqrt{n}} $.
>
> CT-SSF provably outperforms NexCP in terms of average band length, namely, $\mathbb{E} [ H(Q_{1-\alpha}(V^f_{D_{ca}}), X_{test}) ] < Q_{1-\alpha}(V^o_{D_{ca}})$, where the expectation is taken over the calibration fold and the testing point $ (X_{test}, Y_{test}) $.
>
> The reduction in interval widths is primarily attributed to the use of latent features, which enable a more precise capture of the underlying data structure. Different from most existing conformal prediction algorithms, which regard the base model as a black-box model, feature-level operations allow seeing the training process via the trained feature. For a well-trained base model, feature-level techniques improve efficiency by utilizing the powerful feature embedding abilities of well-trained neural networks.
>
> The cubic conditions assume the latent space has a smaller distance between individual non-conformity scores and their quantiles, which reduces the cost of the quantile operation. We provide experimental results in Table 2 in the attached PDF comparing the average distance between each sample and their quantile in the latent space. The results show that the feature space has a smaller distance between individual non-conformity scores and their quantiles, which is key to improving prediction efficiency. Additionally, Figure 1 in the attached PDF demonstrates that the cubic metric is negatively related to the band length and thus positively related to efficiency. Different configurations of $f$ and $g$ will generate different cubic metrics, leading to variations in the reduction.
>
> Thanks for the great idea of providing an exact bound described in big-O notation for the prediction interval! While deriving the exact bound is challenging due to the inherent variability and complexity of the method (e.g., the structure of the base model, the latent space, the specific implementation of the attention mechanism, as well as the transformation from the latent space to the output space), we will surely consider adding the exact bounds in our future version of this work.
>
> Our method doesn't rely on specific assumptions about the underlying time series, the same as the majority of prior works in CP for time series.

---

### Official Review · Reviewer_aroC · 2024-07-12

**Soundness:** 3
**Presentation:** 3
**Contribution:** 3
**Rating:** 6
**Confidence:** 3

**Summary:**

The paper presents a novel approach called Conformalized Time Series with Semantic Features (CT-SSF) for improving uncertainty quantification in time-series forecasting.
The authors propose leveraging latent semantic features through deep representation learning to dynamically adjust weights for conformal prediction.
This approach aims to address the limitations of existing methods that rely on manually selected weights in the output space.
The paper demonstrates that CT-SSF achieves tighter prediction intervals with valid coverage guarantees, outperforming state-of-the-art methods in both synthetic and real-world datasets.

**Strengths:**

S1. The paper introduces an interesting combination of conformal prediction and deep representation learning, focusing on the latent space rather than the output space.

S2. This paper provides theoretical analysis on the validity of the proposed method.

S3. The experimental results demonstrate improvement on the conformal width.

**Weaknesses:**

W1. The latent space is high-dimensional compared to the output space.
Considering high-dimensional statistics is usually much more complicated compared to single-dimensional space, it would be interesting to discuss theoretically or empirically the tightness of the conformal scores in the latent space.

W2. The latent space structure can be very different for different types of architectures.
Hence, the empirical results on different types of deep architectures, e.g., rnn, cnn, transformers, are necessary but missing.

W3. The efficiency of the proposed conformalization method is missing.

**Questions:**

Q1 (cr. W1). Please compare and discuss the impact of switching from output space (single dimension) to the latent space (multiple dimension) on the conformal prediction, e.g., whether and why this may or may not incur an increase in the calibration examples.

Q2 (cr. W2). Please add experimental results for CNN and transformers.

Q3 (cr. W3). Please discuss, report and compare the efficiency.

Q4. In Figure 1, there is no order among the methods, hence a line chart is improper.

**Limitations:**

The author(s) have discussed in the paper.

---

> ### Author Rebuttal · Authors · 2024-08-06
>
> We appreciate your valuable and informative suggestions and would clarify the questions and concerns in the following.
>
> Q1: Please compare and discuss the impact of switching from output space (single dimension) to the latent space (multiple dimension) on the conformal prediction, e.g., whether and why this may or may not incur an increase in the calibration examples.
>
> A1: The main impact of switching CP to the feature space is that in the feature space, it leads to a smaller distance between individual non-conformity scores and their quantiles (i.e., the second argument of the cubic conditions), which is key to improving prediction efficiency in this context. Intuitively, feature space is more semantically rich, providing more fine-grained information to capture the underlying data structure and relationships more accurately, leading to better prediction efficiency. We also provide new experimental results in Table 2 in the attached PDF to compare the average distance between each sample and their quantile in the latent space. The results validate that the distance in feature space is significantly smaller than that in output space, which aligns well with our hypotheses.
>
> Additionally, to further analyze the relationship between the distance in the feature space and the prediction efficiency, we plot the relationship between efficiency (band length) and the cubic metric in Figure 1 in the attached PDF. Specifically, the cubic metric here represents the core statement in the cubic condition (statement 2), which implies a metric form like $\\mathbb{M}[Q_{1-\\alpha}V^f_{D_{ca}} - V^f_{D_{ca}}]$. The results demonstrate that the cubic condition is closely related to the performance of CT-SSF. For each given dataset, the cubic metric  negatively related to the band length and thus positively related to efficiency.
>
> In addition, we conducted additional experiments to analyze the impact of the number of calibration examples on performance, as shown in the following table. Our findings indicate that our method is quite robust to the calibration set size, producing consistently better prediction efficiency. We believe this is because the latent space representations are highly efficient and can capture the essential features of the data well. Consequently, our method does not incur an increase in the calibration examples.
>
> **Table:** The impact of calibration set size on the performance of the evaluated CP algorithms for the real data with CNN as the base model. The specified miscoverage level is $\alpha = 0.1$ for all experiments. The standard deviation is obtained over 10 repeated runs with different random seeds.
>
> |       |         | CT-SSF            | HopCPT            | 100 Calibration  | 200 Calibration  | 300 Calibration  |
> |-------|---------|-------------------|-------------------|------------------|------------------|------------------|
> | **Elec**  | **Cov**   | 90.4±2.20        | 90.2±1.63        | 89.7±3.50        | 89.6±2.11        | 89.6±2.02        |
> |       | **Width** | 0.23±0.04        | 0.32±0.06        | 0.23±0.03        | 0.23±0.03        | 0.23±0.03        |
> | **Stock** | **Cov**   | 91.8±1.82        | 90.6±2.09        | 89.7±2.28        | 90.3±1.70        | 91.1±1.70        |
> |       | **Width** | 0.33±0.08        | 0.53±0.15        | 0.31±0.09        | 0.31±0.08        | 0.32±0.08        |
> | **Weather** | **Cov**   | 89.8±0.27        | 89.9±0.25        | 88.0±0.47        | 89.3±0.32        | 88.8±0.34        |
> |       | **Width** | 0.02±0.005       | 0.04±0.007       | 0.02±0.005       | 0.02±0.005       | 0.02±0.005       |
> | **Wind** | **Cov**   | 88.5±1.83        | 89.2±3.41        | 88.0±2.20        | 88.5±2.50        | 89.0±3.40        |
> |       | **Width** | 0.36±0.05        | 0.60±0.10        | 0.40±0.09        | 0.35±0.10        | 0.40±0.11        |
>
> Q2: Please add experimental results for CNN and transformers.
>
> A2: Thanks for the suggestion. We agree that empirical results on different types of deep architectures are necessary. Experimental results for Transformers have been provided in Table 5, Appendix A.1, which suggest that CT-SSF demonstrates approximately a 10\% reduction in prediction intervals compared to HopCPT across all datasets. Additionally, we conducted new experiments on CNN, and the results can be seen in Table 1 in the attached PDF. We observed that CT-SSF still generates 10\% to 20\% shorter prediction intervals than HopCPT across all datasets.
>
> Q3: Please discuss, report and compare the efficiency.
>
> A3: We compared the prediction efficiency of CT-SSF with other baselines, such as HopCPT and NexCP. Our results, as shown in Tables 1 and 2 of the original paper, indicate that CT-SSF achieves approximately a 5-10\% reduction in average prediction interval length across both synthetic and real-world datasets. In terms of computational efficiency, the runtime varies significantly depending on the size of the datasets. For example, with electricity data, our method takes approximately 40 seconds, while HopCPT takes around 3 minutes, and all other baselines take less than 5 seconds for a single experiment (i.e., one seed, not including training the prediction model). The longer runtime for our method and HopCPT is due to the complexity of the adaptive reweighting and attention mechanisms used, which involve more intensive computations compared to simpler baseline methods.
>
> Q4: In Figure 1, there is no order among the methods, hence a line chart is improper.
>
> A4: We agree with your observation regarding Figure 1. We will remove the line chart and replace it with a scatter plot, to accurately represent the data.
>
> Once again, we appreciate the reviewer's precious time. We are eager to engage in further discussions to clear out any confusion.

---

> > ### Comment · Reviewer_aroC · 2024-08-12
> >
> > Thanks to the authors for addressing my concerns and providing additional experimental results. I remain positive about this work with my previous score.

---

### Official Review · Reviewer_phaJ · 2024-07-13

**Soundness:** 3
**Presentation:** 2
**Contribution:** 2
**Rating:** 6
**Confidence:** 3

**Summary:**

This paper proposes Conformalized Time Series with Semantic Features (CT-SSF), a conformal prediction method that constructs prediction intervals for time series data using nonconformity scores computed in the latent feature space of neural networks. CT-SSF further assigns time-dependent weights to the scores based on their similarity to the current prediction errors.

**Strengths:**

The idea of adapting time-dependent weights for the nonconformity score computed in semantic feature space is novel.

The authors conducted experiments on both synthetic and real datasets to demonstrate that, on average, the width of the prediction interval is shorter than those obtained from existing approaches. Additionally, the authors conducted an ablation study to investigate the performance of each core component in the CT-SSF method.

**Weaknesses:**

Clarity: The clarity of the writing can be improved. For example: In the experimental setup (Section 5.1), it is mentioned that CT-SSF is model-agnostic, allowing any prediction models for time series to be used as the base models. In contrast, the conclusion states that one of the limitations of CT-SSF is its reliance on NNs, excluding simpler models like ridge regression or random forest. The clarity of the paper would benefit from more rigorous writing, such as restricting the first statement to 'any NN-based prediction models for time series.'

Experimental Significance: The standard errors in the synthetic data experiments are based on only five repetitive experiments. As a result, the confidence intervals on widths are too wide to demonstrate significant advantages of the CT-SSF method. The experimental results would be more convincing if more repetitive experiments were conducted with additional seeds.

Theoretical contribution and unique challenges: As the authors have acknowledged, the theorem and its proof in this paper are identical to Theorem 4 in Teng et al. (2022). Therefore, the unique challenges faced in this project are not clearly articulated. Please refer to question number 1 for further clarifications.

**Questions:**

I have several clarifying questions regarding the methodology and experiments outlined in the paper:

Methodology:
If my understanding is correct, CTSSF is an incremental methodology based on Nex-CP (using weighted conformal prediction), FeatureCP (implementing CP within a latent feature space instead of in the output space), and HopCPT (using the same intuition of designing weights according to how similar previous errors are to the current error). I’m confused about the last part—how exactly do you update the weight? Did you rely on the MHN attention mechanism as in the HopCPT paper? Please correct me if I misunderstood or overlooked anything - I would appreciate it if the authors could highlight the unique challenges faced in this project. If the authors could clarify these challenges and provide experimental results with a larger number of independent repetitions to enhance the significance of their results (please see 'Experimental Significance' under weaknesses), I would be happy to increase my score.

Experiment:
Did you compare your results with other benchmarks used for conformal time series forecasting, such as the EnbPI [1] method and the SPCI [2] method?

How many seeds did you use for real data experiments?

Reference:
[1]: “Conformal Prediction Interval for Dynamic Time-Series.” Chen Xu and Yao Xie, PMLR, 2021
[2]: “Sequential Predictive Conformal Inference for Time Series.” Chen Xu and Yao Xie, PLMR, 2023

**Limitations:**

The author addressed the limitations of their work.

---

> ### Author Rebuttal · Authors · 2024-08-06
>
> Thank you for your insightful feedback on our manuscript. We appreciate your comments and have taken them into consideration to improve our paper. Below, we do our best to address the reviewer's questions adequately such that we could receive a better score.
>
> Q1: Comparisons with prior works and unique challenges.
>
> A1: Our work is indeed greatly inspired by Nex-CP, FeatureCP, and HopCPT. However, CT-SSF is not incrementally a combination of the three. A simple solution would be to combine Nex-CP with FeatureCP. However, Nex-CP employs pre-defined weights instead of data-adaptive weights, which prevents it from fully capturing the intrinsic characteristics of the original time series. While Nex-CP uses weighted conformal prediction and FeatureCP operates within a latent feature space, CT-SSF uniquely integrates these concepts by applying adaptive reweighting directly within the latent feature space. This integration allows for more precise capturing of the underlying data structure, leading to tighter prediction intervals and improved prediction accuracy.
>
> Compared to HopCPT, our weight update mechanism uses a standard simple attention weights mechanism rather than the MHN attention mechanism from the HopCPT paper. We chose the standard attention mechanism for its simplicity and computational efficiency, which still allows for dynamic adjustment of data point significance. Despite using a typical attention mechanism, our results demonstrate that CT-SSF outperforms HopCPT in both performance and computational efficiency because the attetion-based mechanism effectively prioritizes relevant data points and a simple standard attention mechanism can reduce computational overhead. The simplicity and efficiency of the standard attention mechanism enable faster computations and more scalable implementations without sacrificing accuracy.
>
> The primary unique challenge we faced in this project was ensuring the effective integration of attention-based weight updates within the latent feature space. Achieving this required innovative solutions to ensure that the dynamic adjustments made by the attention mechanism did not introduce significant distortions or inconsistencies in the prediction intervals. To address this, we employed continuity-preserving feature extraction methods, such as RNNs, to maintain the intrinsic structure of the data. Additionally, we utilized Transformer models for dynamically adjusting weights, which allowed us to effectively prioritize data points based on their relevance to the current prediction task.
>
> Q2: Experimental Significance.
>
> A2: Thanks for the suggestion.
> We did not include comparisons with the EnbPI and SPCI benchmarks in our current study because HopCPT was shown to significantly outperform these methods in the original paper. However, we understand the importance of comprehensive benchmarking. Therefore, we have conducted experiments on EnbPI and SPCI with 10 repetitive trials, and the results are presented in Table 1 in the attached PDF. Our findings indicate that CT-SSF demonstrates a 10\%-20\% reduction in prediction intervals compared to the SOTA HopCPT across all datasets, and over a 20\% reduction for SPCI and EnbPI.
>
> For the real data experiments, we used five different seeds to ensure the robustness of our results. We agree that using more seeds could further strengthen our findings. Therefore, we increased the number of repetitions to 10 for the new results in Table 1 in the attached PDF. We see that variance is reduced as expected and CT-SSF still outperforms all other baselines.
>
> Q3: Clarity.
>
> A3: Thank you for pointing this out. We acknowledge the inconsistency in our statements regarding the model-agnostic nature of CT-SSF. We have revised the statement in Section 5.1 to specify that CT-SSF is compatible with any NN-based prediction models for time series. The revision is below:
> "CT-SSF is model agnostic, therefore, any NN-based prediction models for time series prediction models can be used as the base models. To better show the advantage of our proposed method, we utilize a Recurrent Neural Network (RNN) model, which can be replaced with more advanced models like Transformers."
>
> Once again, we appreciate the reviewer's precious time. We are eager to engage in further discussions to clear out any confusion.

---

> > ### Comment · Reviewer_phaJ · 2024-08-10
> >
> > Thank you to the authors for addressing my comments and conducting additional experiments. The responses and new results have addressed my concerns regarding experimental significance and unique challenges. I have increased my score accordingly.

---

### Official Review · Reviewer_iFYG · 2024-07-15

**Soundness:** 3
**Presentation:** 3
**Contribution:** 3
**Rating:** 6
**Confidence:** 3

**Summary:**

The paper presents a conformal prediction approach for time series data in the latent space of the prediction model – Conformalized Time Series with Semantic Features (CT-SSF). Non-conformity scores constructed in the latent space are expected to capture deeper insights of the data and temporal dynamics and improve prediction efficiency. The paper also proposes an adaptive weight adjustment scheme in the latent space to dynamically adjust weights such that larger weights are assigned to more relevant data points. Authors perform experiments on synthetic and real-world datasets to demonstrate the effectiveness of their proposed method compared to existing conformal prediction methods for distribution drift and time series.

**Strengths:**

**Originality**: The paper extends HopCPT [1], a conformal prediction method for time series data, by constructing non-conformity scores in the latent space using ideas and methods from [2] (e.g., band estimation). The main original contribution in my opinion is the weighted gradient mechanism. While CT-SSF combines methods from [1] and [2] for the most part, this novel combination is effective and demonstrates performance improvement over existing methods.


**Quality**: The paper is technically sound. The claims are well supported by theory and empirical evaluation over multiple datasets and comparison with baselines. The paper also includes ablation studies to demonstrate the effectiveness of the method.


**Clarity**: The paper is well-written, organized, and easy to follow for the most part.

**Significance**: The proposed method constructing non-conformity scores in the latent space provably outperforms conformal prediction methods in the output space (e.g., NexCP) in terms of average prediction interval length under the stated assumptions. The empirical analysis also demonstrates consistent results. I believe these results will be of interest to the uncertainty quantification as well as broader community.


**References**

[1] Andreas Auer, Martin Gauch, Daniel Klotz, and Sepp Hochreiter. Conformal prediction for time series with modern hopfield networks. Advances in Neural Information Processing Systems, 36, 2023.

[2] Jiaye Teng, Chuan Wen, Dinghuai Zhang, Yoshua Bengio, Yang Gao, and Yang Yuan. Predictive inference with feature conformal prediction. arXiv preprint arXiv:2210.00173, 2022.

**Weaknesses:**

1. Limited discussion of past work: While the paper largely borrows ideas from [1, 2], I feel the text lacks appropriate context for complete understanding. There is limited explanation of the band estimation technique as well as HopCPT. I understand the main paper might not have space but this can be added to the appendix.

2. There is less clarity on what the assumptions for Theorem 1 rely on and how they translate in practice. It would be helpful to discuss the robustness of the assumptions and implications if they are violated (in theory or practice) since this is the main result of the paper.


**References**

[1] Andreas Auer, Martin Gauch, Daniel Klotz, and Sepp Hochreiter. Conformal prediction for time series with modern hopfield networks. Advances in Neural Information Processing Systems, 36, 2023.

[2] Jiaye Teng, Chuan Wen, Dinghuai Zhang, Yoshua Bengio, Yang Gao, and Yang Yuan. Predictive inference with feature conformal prediction. arXiv preprint arXiv:2210.00173, 2022.

**Questions:**

1. Can the authors elaborate on how $\tilde{w}$ is updated in Algorithm 1?

2. Minor typo: p6 l243 -> calibration

**Limitations:**

The authors discuss limitations in the Limitations section. Authors briefly discuss the potential social impact in A.3.

---

> ### Author Rebuttal · Authors · 2024-08-06
>
> We extend our sincere appreciation for your valuable feedback and suggestions. Regarding your concerns, we would like to offer further clarification.
>
> Q1: Can the authors elaborate on how $\tilde{w}$ is updated in Algorithm 1?
>
> A1: The weights $\tilde{w}$ are updated based on the attention mechanism of the Transformer model. The weights are adjusted dynamically to reflect the significance of each data point in the prediction task.
> The attention weights from the Transformer model are used to assign higher importance to data points that have more influence on the prediction:
> $\tilde{w} \leftarrow \text{AttentionWeights}(\hat{g}(u), Y).$
> Here, $\text{AttentionWeights}(\hat{g}(u), Y)$ represents the attention weights calculated by the Transformer model, reflecting the similarity between the predicted output $\hat{g}(u)$ and $Y$. Finally, we adjust the latent vector $u$ using the weighted gradient descent mechanism. This step ensures that the updates to $u$ are influenced by the dynamically adjusted weights $\tilde{w}$, focusing the learning process on the most critical data points. We will add these explanations in the future version of this work.
>
> Q2: Limited discussion of past work.
>
> A2:
> Thanks for the suggestion. We acknowledge that our initial submission did not provide a comprehensive discussion of the previous works due to the page limit.
>
> To remedy this, we have added more detailed descriptions of current CP approaches for time series data in the appendix, which is also attached below.
>
> There are generally three primary approaches designed to manage these challenges and enhance the reliability and validity of the CP in time series: reweighting, updating non-conformity scores and updating significance level.
>
> Reweighting assigns relevance-based weights to data points to align the data distribution closer to a target distribution. NexCP uses exponentially decayed weights to emphasize recent observations, but these lack adaptability. HopCPT improves on this by using a Modern Hopfield Network (MHN) for similarity-based reweighting. It assigns weights to past time steps based on their relevance to the current time step. Encoded inputs are processed with learned weight matrices, and a hyperparameter adjusts the focus of the probabilistic distribution. These weights create weighted conformal prediction intervals by discounting extremal quantiles of relative errors.
>
> The second technique, updating non-conformity scores leverages the most recent $T$ data points and continuously updates prediction intervals as new data becomes available. For example, EnbPI uniquely updates the non-conformity score with sequential error patterns to adapt the intervals dynamically. And SPCI replaces the empirical quantile with an estimate by a conditional quantile estimator to effectively address serial dependencies among residuals in sequential analysis.
>
> The last main direction for CP in time series focuses on adaptively adjusting the significance level $\alpha$ during test time to account for mis-coverage. This method can dynamically adjust the size of prediction sets in an online setting where the data generating distribution is allowed to vary over time in an unknown fashion.
> For example, the update rule for the quantile level $ \alpha $ in ACI is: $\alpha_{t+1} = \alpha_t + \gamma(\alpha - \text{err}_t)$,
> where $ \gamma $ is a step size parameter, and $ \text{err}_t $ indicates if $ Y_t $ was not included in $ \hat{C}_t(\alpha_t) $. The approach ensures that the prediction intervals adjust over time to account for shifts in the data distribution, maintaining the desired coverage probability.
>
> Q3: There is less clarity on what the assumptions for Theorem 1 rely on and how they translate in practice. It would be helpful to discuss the robustness of the assumptions and implications if they are violated (in theory or practice) since this is the main result of the paper.
>
> A3: We understand the importance of clearly discussing the assumptions underlying Theorem 1 and their practical implications. Below, we have provided an enhanced explanation to address the robustness of the assumptions and the potential consequences if they are violated.
>
> Assumption 1, Length Preservation in the Feature Space, ensures that the transformation from the feature space to the output space maintains consistent interval lengths. This assumption is generally robust with smooth and continuous feature extraction methods, but abrupt changes can lead to inconsistent intervals, affecting prediction reliability. Assumption 2, Expansion by Band Estimation Operator, captures differences in interval lengths and quantiles. It is robust if the operator is well-calibrated, though sensitive to calibration data quality and quantity. Violations may result in inaccurate prediction intervals, compromising coverage guarantees. This assumption directly influences the performance of our method, so we conduct new experiments to validate this assumption in Table 2 in the attached PDF. Results show that the latent space has a smaller distance between individual non-conformity scores and their quantiles, which reduces the cost of the quantile operation. Finally, Assumption 3, Quantile Stability, ensures stable band lengths across feature and output spaces, crucial for consistent coverage. This assumption is typically robust with well-behaved data distributions but can be challenged by extreme values, leading to unstable intervals and undermining coverage validity. To mitigate these risks, we recommend using continuity-preserving feature extraction methods, regularly recalibrating the band estimation operator with diverse data, and employing robust statistical methods for outlier detection. We will add these explanations in the future version of this work.

---

> ### Comment · Reviewer_iFYG · 2024-08-11
>
> I thank the authors for addressing the questions and look forward to future versions with more detailed discussion as suggested. I remain positive of the work as earlier.

---

### Author Rebuttal · Authors · 2024-08-06

We would like to thank all reviewers for their thorough reviews, helpful suggestions, and constructive comments. To summarize, we made the following main changes to the manuscript to follow the suggestions and comments of the reviewers:
1. To illustrate the generalizability and performance of CT-SSF, we have added additional experimental results using CNN as the base model with more repetitions. Results in Table 1 in the attached PDF suggest a 10\% to 20\% reduction in prediction intervals compared to CP for time series benchmarks and standard CP.
2. We conducted additional experiments to bridge the gap between theory and empirical results. Table 2 in the attached PDF compares the quantile and non-conformity scores in the latent space and output space, suggesting that the latent space has a smaller distance. Experimental results in Figure 1 in the attached PDF indicate that prediction efficiency is positively related to the cubic metric. These results offer enhanced insights into the theoretical framework.
3. Recognizing the importance of discussing past work, we have added detailed descriptions of CP for time series in the appendix.
4. We revised typo errors and inconsistencies, and removed the line chart in the ablation study.

---

### Decision · Program_Chairs · 2024-09-25

**Decision:**

Accept (poster)

**Comment:**

The paper studies conformal prediction in the context of time series prediction problems. After the rebuttal and discussion phase there is a clear convergence of (the majority) of reviewers (who engaged in discussion) towards a positive impression of the paper (in terms of soundness, experimental validation and clarity). Considering the reviewer comments and the rebuttal, and including the score increases, I am recommending acceptance at this point, but I am also encouraging the authors to take all comments seriously and adjust the manuscript accordingly.